# Parental Hesitancy towards the Established Childhood Vaccination Programmes in the COVID-19 Era: Assessing the Drivers of a Challenging Public Health Concern

**DOI:** 10.3390/vaccines10050814

**Published:** 2022-05-20

**Authors:** Christos Derdemezis, Georgios Markozannes, Marina O. Rontogianni, Marianthi Trigki, Afroditi Kanellopoulou, Dimitris Papamichail, Eleni Aretouli, Evangelia Ntzani, Konstantinos K. Tsilidis

**Affiliations:** 1Department of Hygiene and Epidemiology, School of Medicine, University of Ioannina, 45110 Ioannina, Greece; cderdemezis@yahoo.gr (C.D.); georgemarkozannes@gmail.com (G.M.); marinaront@gmail.com (M.O.R.); marianthitrigki@gmail.com (M.T.); afkanellopoulou@gmail.com (A.K.); entzani@uoi.gr (E.N.); 2Department of Public Health Policy, University of West Attica, 11521 Athens, Greece; dpapamnos@gmail.com; 3Laboratory of Cognitive Neuroscience, School of Psychology, Aristotle University of Thessaloniki, 54124 Thessaloniki, Greece; eleni.aretouli@gmail.com; 4School of the Social Sciences, University of Ioannina, 45110 Ioannina, Greece; 5Center for Evidence Synthesis in Health, Department of Health Services, Policy, and Practice, School of Public Health, Brown University, Providence, RI 02912, USA; 6Institute of Biosciences, University Research Center of Ioannina, University of Ioannina, 45110 Ioannina, Greece; 7Department of Epidemiology and Biostatistics, School of Public Health, Imperial College London, London W2 1PG, UK

**Keywords:** vaccine hesitancy, childhood vaccination, COVID-19, Greece

## Abstract

(1) Background: Vaccine hesitancy remains a major public health concern. The reasons behind this attitude are complex and warrant careful consideration, especially in the context of the COVID-19 era. The purpose of this study was to estimate vaccine hesitancy towards the established childhood immunization programmes in a non-random sample of Greek parents and explore possible links with important drivers of this phenomenon. (2) Methods: An online self-administered questionnaire was used from October 2020 to April 2021 to collect socio-demographic, lifestyle, and health status data and evaluate knowledge, views, and attitudes of the Greek population on COVID-19 pandemic-related issues. Parents were further asked to complete the Parent Attitudes about Childhood Vaccines (PACV) questionnaire. (3) Results: A total of 1095 parents participated in the study with a mean age of 50 years (SD 9.5 years). The hesitancy against the established childhood vaccinations was estimated at 8.9% (95% CI, 7.3–10.8%). Married status and higher education and income were negatively correlated with hesitancy, whereas positive correlations were found for stress and depressive symptoms and current smoking. Variables related to proper awareness, sound knowledge, and trust toward authorities regarding the COVID-19 pandemic were strongly associated with being less hesitant against the established childhood vaccination programmes. (4) Conclusion: The estimated parental hesitancy against the established childhood vaccination programmes is worrisome. Variables related to good awareness and knowledge of the COVID-19 pandemic were strongly associated with being less hesitant against childhood vaccinations. Since controversy surrounding COVID-19 vaccinations may decrease parents’ confidence in routine childhood vaccinations, appreciating the complex reasons behind vaccine hesitancy may inform public health policies to overcome barriers and increase vaccine acceptance.

## 1. Introduction

Ever since the administration of the first vaccine against smallpox over 200 years ago, the development of vaccines to combat communicable (and more recently, non-communicable) diseases has been a major thrust in disease prevention and increase in life expectancy. Childhood vaccinations are considered one of the most significant achievements of public health. According to estimates from the World Health Organization (WHO), each year, childhood vaccinations prevent 2–3 million deaths [1]. However, adherence to childhood vaccination programmes still falls far from the desired 100%, despite the often-mandatory nature and generally free distribution and administration. Vaccine hesitancy, defined as the reluctance or refusal to vaccinate despite the availability of vaccines [2], was considered by WHO as one of the top ten threats to public health worldwide in 2019 [3]. Vaccine hesitancy varies widely depending on how it is assessed, the geographic region, and/or whether it addresses all or a specific vaccine [4,5]. Recent reports have identified drivers of vaccine confidence in many countries, revealing also that a gap in vaccine trust remains around the globe [4,5]. Some of the major factors associated with vaccine hesitancy are younger age, male gender, lower socioeconomic status and education along with lower trust in vaccine importance, safety, and effectiveness, and lower trust in health authorities and government. Carefully addressing the complex reasons leading to this attitude can enable us to better inform public health policies, overcome barriers, and increase vaccine acceptance and administration rates [6].

Along with vaccine hesitancy, high-threat pathogens were also included in the top ten threats to public health [3]. Previous years have seen the spread of epidemic pathogens such as Ebola or Middle East respiratory syndrome (MERS) coronavirus and the need to prepare for unknown future pathogens was emphasized [7]. In late 2019, the world witnessed the spread of a novel severe acute respiratory syndrome coronavirus (SARS-CoV-2) that quickly escalated to a pandemic by early 2020. As of February 2022, there have been more than 458 million cases and 6 million deaths worldwide [8,9].

Vaccines against coronavirus disease 2019 (COVID-19) were developed and tested in 2020 and administration begun by late 2020 or early 2021. Limits in upscaling their production and thus initial shortage of vaccines was soon followed by skepticism from a proportion of the public over efficacy and safety, despite positive authorization by respective health authorities. The spread of misinformation about the benefits and risks of COVID-19 vaccines further worsened the public’s hesitancy [10], reaching figures as high as 50% [11]. To date, data from a limited number of studies have indicated that vaccine hesitancy against SARS-CoV-2 and the COVID-19 pandemic may lead to a drop in the rates of the established childhood vaccine programmes [12,13,14,15]. We aimed to estimate vaccine hesitancy toward the established childhood immunization programmes in a non-random sample of parents living in Greece during the COVID-19 pandemic era, and gain insight into the possible determinants of vaccine hesitancy.

## 2. Methods

The paper was written in accordance with the Strengthening the Reporting of Observational Studies in Epidemiology (STROBE) guidelines [16].

We utilized an online self-administered questionnaire consisting of 50 closed-ended questions to evaluate socio-demographic, lifestyle, and health status characteristics, the knowledge, views, and attitudes of the Greek population on COVID-19 pandemic-related issues, and parental vaccine hesitancy toward the established childhood immunization programmes. It was administered from October 2020 to April 2021 through different online platforms, mainstream social media, and local press. In addition, participants of a population-based cohort study in northwestern Greece, the Epirus Health Study (EHS) [17,18,19], were also invited to complete the online survey. To guarantee anonymity, questions about personal data were avoided. Ethics approval was obtained from the Ethics Committee at the University of Ioannina.

### 2.1. Data Collection

#### Socio-Demographic Characteristics, Lifestyle Factors, and General Health Status

The questionnaire included information about (I) socio-demographic characteristics, such as age, sex, marital status, number of children, educational level, employment status, income, and geographical area of residency (assessed by postcode); (II) lifestyle factors, including smoking status, alcohol consumption, recreational physical activity, and self-reported height and weight; (III) data regarding general health status and medical history, including self-reported presence of chronic diseases, physician diagnosis of diabetes mellitus, hypercholesterolemia and hypertension, self-perceived stress or depressive symptoms within the last two weeks, and weight change within the last 6 months of completing the questionnaire.

### 2.2. COVID-19 Related Variables

A set of 15 close-ended questions was used to evaluate participants’ knowledge, views, attitudes, and trust of officials regarding the COVID-19 pandemic, their compliance with public health mitigation measures, and willingness to vaccinate against SARS-CoV-2. A detailed description of these variables is available in Appendix A.

### 2.3. Assessment of Childhood Vaccine Hesitancy

Participants reporting being parents to at least one child were further asked to complete the Parent Attitudes about Childhood Vaccines (PACV) questionnaire [20]. PACV is a validated, self-administered questionnaire that contains 15 items about behavior, safety and efficacy, and general attitudes toward childhood vaccine hesitancy. The outcome of interest in this study was vaccination hesitancy as assessed by the PACV. Each answer was granted points resulting in a score, which was then converted to a percentage scale. Cut-off points were set at 50 and 70 points (score 0–50: not hesitant, 51–70: possibly hesitant, and 71–100: probably hesitant). Due to the low number of participants in the last category in our sample, we chose to use the dichotomous variable “not hesitant” (score 0–50) vs. “hesitant” (score 51–100) as the dependent variable.

Furthermore, we aimed to assess the repeatability of the response to the PACV questionnaire and evaluate factors that can lead to potential changes in childhood vaccine hesitancy over time. Specifically, a subset of the responders of the online questionnaire, who were participants of the EHS or had expressed interest in taking part in the EHS, were invited to complete the PACV questionnaire one more time during March and April 2021.

### 2.4. Statistical Analysis

Participants’ socio-demographic, lifestyle, and health-related characteristics, and the COVID-19-related variables were investigated as potential determinants of childhood vaccine hesitancy. We excluded variables assessing whether participants were diagnosed with COVID-19 because of the extremely low number of participants who answered these questions positively. The descriptive results of the aforementioned factors were presented overall and according to sex and childhood vaccine hesitancy status using means and standard deviations (SD) for continuous variables, and counts and percentages for categorical variables. Differences among subgroups were examined using the chi-squared test for categorical variables and independent samples *t*-test for continuous variables. The prevalence of childhood vaccine hesitancy was estimated assuming a binomial distribution, after standardizing for the corresponding age distribution of the overall Greek population using data from the Hellenic Statistical Authority [9].

Two nested logistic regression models were developed to examine the independent association of the COVID-19-related variables and childhood vaccine hesitancy, namely, a minimally adjusted model for continuous age, sex, education, and monthly income, and a maximally adjusted model further adjusting for presence of depressive symptoms during the last 2 weeks, recreational physical activity, profession, health status, smoking status, and continuous body mass index (BMI). A detailed description of the adjusted variables is available in Appendix A. As a sensitivity analysis, we categorized the participants by geographic regions of residency and fitted the minimally and maximally adjusted models using these regions as clusters to account for possible regional effects. The Benjamini–Hochberg false discovery rate (FDR) procedure was performed for the adjustment for multiple comparisons [21]. The analysis was also performed using the continuous vaccine hesitancy score as our dependent variable in linear regression models, to evaluate the robustness of our results.

Finally, to assess the repeatability of the PACV questionnaire in the subset of participants with a second assessment, we computed the intraclass correlation coefficient (ICC) based on a two-way random effects model. We used linear regression models to evaluate the association of the COVID-19-related variables with the difference in PACV scores.

Statistical analyses were performed using STATA (version 14; StataCorp, College Station, TX, USA).

## 3. Results

### 3.1. Description of Participants

Of the 1754 individuals that completed the online questionnaire, a total of 1095 participants (380 men, 34.7%) reported being parents to at least one child and answered the PACV questionnaire, which comprised our analytical sample size (Table 1). Their age ranged from 18 to 75 years with a mean of 50.3 (SD = 9.4) years; 928 (85.7%) were married or in a cohabitation agreement, and the mean BMI was 26.3 (SD = 4.7) kg/m [2]. Eight-hundred-and-twelve (74.5%) participants had completed higher education (university degree or above), and 54.8% of participants had a monthly income of at least EUR 1101 (N = 542). A total of 421 participants (38.6%) were current smokers and 31.9% consumed alcohol at least weekly (N = 349). Presence of chronic disease was reported by 36.5% of participants (N = 392), and good or very good health status by 86% (N = 939).

The majority of socio-demographic and lifestyle characteristics differed significantly by sex (Appendix A), with male participants being older, with higher income, and more likely to be employed in manual labor jobs; they had higher BMI but were more physically active, and they were less likely to be current smokers and more likely to drink alcohol compared to women (all *p*-values ≤ 0.001). Stress or depression symptoms during the previous two weeks were more frequent among women (*p*-values < 0.001), while education, presence of chronic diseases, health status, and diagnosis of diabetes or hypercholesterolemia did not differ significantly by sex.

Most of the COVID-19-related variables did not differ by sex except for women reporting higher compliance to the COVID-19 public health mitigation measures more frequently compared to men (*p*-values < 0.001), and men being more willing to vaccinate themselves or their children against coronavirus (*p*-values < 0.001) (Appendix A).

### 3.2. Prevalence and General Determinants of Vaccine Hesitancy

The mean PACV score was 20.4 (SD = 18.9) points across the 1095 parents, and there was a slight non-linear increasing trend (*p*-value of increase = 0.006) in the mean score with time from October 2020 to April 2021 (Figure 1). The prevalence of parental vaccine hesitancy defined as a PACV score of more than 50 points was 8.9% (95% CI: 7.3, 10.8, N = 98) after combining the possibly (51–70; 6.3%) and probably hesitant (71–100; 2.6%) categories. The standardized prevalence of vaccine hesitancy by age, using the stratified Greek population as the standardization data, was 9.1%.

The prevalence of parental vaccine hesitancy did not differ by sex (Appendix A), age (9.1% in parents younger than 50 vs. 8.8% in parents 50 years or older; *p*-value = 0.89), BMI, or alcohol consumption (Table 1), but it varied between participants residing in different geographical regions, ranging from 6.2% among Epirus residents to 16.7% for residents of Peloponnese (*p*-value = 0.006) (Table 1). In addition, non-hesitant participants were more likely to be married or in a cohabitation agreement, have higher education, have executive job positions, and report a higher income (all *p*-values ≤ 0.001). Conversely, the prevalence of stress and depressive symptoms and current smoking was higher among hesitant participants (*p*-values ≤ 0.01).

### 3.3. COVID-19-Related Determinants of Vaccine Hesitancy

Most of the COVID-19-related variables were associated with childhood vaccine hesitancy. In the logistic regression models adjusted for age, sex, education, and income (Table 2), parents believing in coronavirus’ existence, having good knowledge about the pandemic, following COVID-19 mitigation public health measures, trusting health authorities, government, and official information sources, being vaccinated against the flu and willing to vaccinate themselves or their children against SARS-CoV-2, and reporting lower capability to protect themselves against COVID-19 were all strongly associated with being less hesitant with respect to the established childhood vaccination programmes.

Based on the maximally adjusted logistic regression models (Table 2), participants who believed that the coronavirus probably or definitely exists had 80% (OR = 0.20, 95% CI: 0.05–0.82) and 95% (OR = 0.05, 95% CI: 0.01–0.19) lower odds of being hesitant, respectively, compared to the participants who did not believe in the existence of the coronavirus. Likewise, compliance with public health measures of personal protection (almost always vs. never/rarely; OR = 0.09, 95% CI: 0.03–0.22) and public protection (OR = 0.06, 95% CI: 0.02–0.17) for COVID-19 mitigation were inversely associated with childhood vaccine hesitancy. Inverse associations were also observed between childhood vaccine hesitancy and trust in health authorities (absolute vs. no trust; OR = 0.04, 95% CI: 0.01–0.19), official information sources (OR = 0.08, 95% CI: 0.03–0.23), and the government (OR = 0.05, 95% CI: 0.01–0.35) to mitigate the pandemic. Compared to participants not vaccinated for seasonal flu, vaccinated participants during the present and previous year had 92% (OR = 0.08, 95% CI: 0.03–0.21) and 82% (OR = 0.18, 95% CI: 0.07–0.46) lower odds of being hesitant, respectively. Finally, an inverse association was also observed between willingness (vs. not) to vaccinate their children against coronavirus with childhood vaccination hesitancy (OR = 0.02, 95% CI: 0.00–0.12). The results did not differ after the FDR correction for multiple comparisons.

When residential postcode was further adjusted for in the multivariable logistic regression models as a cluster variable, the results were extremely similar (Appendix A). Analysis with the vaccine hesitancy score as a continuous variable, using linear regression models, did not find qualitative differences compared to the results from the logistic regression models (results not shown).

### 3.4. Vaccine Hesitancy in the Repeated Assessment

Out of 289 participants that were re-contacted, 156 (54%) answered the PACV questionnaire again between March and April 2021. The time period between the two assessments ranged from 25 to 167 days with a median of 113 (IQR: 90–159) days. The ICC between the two measurements was 0.64 (95% CI: 0.53, 0.72) (Figure 2). We observed no difference regarding childhood vaccine hesitancy in 151 participants (149 were non-hesitant and 2 were hesitant), whereas 4 hesitant participants were classified as non-hesitant in the second assessment and 1 non-hesitant became hesitant. We did not observe any association between the COVID-19-related variables measured at the time of the first assessment and change in the childhood vaccine hesitancy (Appendix A).

## 4. Discussion

In our non-random sample of 1095 Greek parents, hesitancy against the established childhood vaccinations was assessed using the PACV questionnaire from October 2020 to April 2021 and estimated at 8.9%. Married participants with higher education and income, non-smokers, and participants with less stress and depressive symptoms were less likely to be hesitant. Variables related to good awareness, knowledge, and trust in authorities toward the COVID-19 pandemic were strongly and independently associated with less hesitancy against the established childhood vaccination programmes.

Vaccine acceptance rates have generally increased within the last few decades both in developed and in developing countries [4,5,22], as also suggested by higher vaccination coverage globally [23]. However, there have been exceptions where misinformation has driven public trust into casting serious doubts regarding vaccine safety and efficacy, an example being measles outbreaks related to lower vaccination rates [24]. Vaccine hesitancy, being complex and context-specific [2], may vary depending on the assessment method [22,25]; therefore, we chose the PACV questionnaire as the only measure available that we know of that is explicitly designed to identify parental vaccine hesitancy [20]. In addition, PACV has been shown to have good construct and predictive validity and reliability [20,26,27]. Our findings are comparable to the findings of studies conducted in other countries using PACV reporting vaccine hesitancy ranging from as low as 5.9% to 20% or higher [28,29,30,31,32,33,34,35].

Furthermore, our finding of 8.9% of childhood vaccine hesitancy seems to be in accordance with other recent data from Greece. In a large-scale retrospective analysis examining global trends in vaccine confidence, 94% of Greek participants responded that vaccines are important for children [5]. Another recent study including data derived from 1657 school-aged children investigated vaccine coverage between 2016–2019 in western Greece and showed generally high vaccination coverage rates (>90%) for the majority of vaccines [17], but also reported suboptimal coverage for a number of vaccines such as the pneumococcal conjugate, the human papillomavirus, and the meningococcal serogroup B vaccines. A large and representative study published in 2010 reported similarly high (77.1% to 98.3% depending on specific vaccine) but not optimal vaccination rates [36].

Identifying which factors are associated with vaccine hesitancy is the first necessary step to decipher how to intensify and tailor public health strategies and policies to mitigate vaccine hesitancy. Several studies and reviews have addressed this issue and identified determinants of vaccine hesitancy [4,5,22]. Complex interactions between socio-demographic factors, access to information and misinformation, and trust in government and health authorities are known risk factors. Our study supports prior evidence that socio-economic factors such as marital status, job type, and higher education are positively associated with vaccination rates [36,37].

The identification of factors associated with childhood vaccine hesitancy may be further complicated amidst the COVID-19 pandemic, especially with regard to the information and misinformation on SARS-CoV-2 vaccines [37]. Willingness to vaccinate was previously found to be consistently associated with trust in health professionals, whereas disbelief and mistrust of health professionals and government may undermine vaccination policies and rates [38]. We addressed whether awareness and knowledge regarding the COVID-19 pandemic and trust in authorities to mitigate it may impact childhood vaccine hesitancy. Our findings suggest that poor awareness and knowledge and limited trust in authorities regarding COVID-19 are strongly associated with childhood vaccine hesitancy in the COVID-19 era.

To the best of our knowledge, our study was among the first to examine the association between childhood vaccine hesitancy and COVID-19-related variables including willingness to vaccinate against SARS-CoV-2. Our research covered a critical period before and during the first phases of SARS-CoV-2 vaccine distribution in Greece, and attempts were made to examine change in vaccine hesitancy rates in our population over time, but the seven-month duration of the study was not long enough to capture large changes. There are also other limitations that need to be acknowledged. Our sample was not random or representative of the Greek population, limiting the applicability of the prevalence of childhood vaccine hesitancy and generalizability of our results. However, when we standardized the vaccine hesitancy by age and accounted for possible regional effects in our association estimates, the findings did not change.

## 5. Conclusions

In this cross-sectional study of 1095 Greek parents, we estimated vaccine hesitancy toward the established childhood immunization programmes at 8.9%, which can be considered worrisome given the high efficacy and almost non-existing safety concerns of childhood vaccines. Variables related to good awareness, knowledge, and trust toward the COVID-19 pandemic were strongly associated with being less hesitant. Appreciating the complex reasons behind vaccine hesitancy may inform public health policies to overcome barriers and increase vaccine acceptance and trust. We should not allow inaccurate information about the COVID-19 pandemic and SARS-CoV-2 vaccines to backfire on parents’ entrustment of childhood immunizations. Future efforts should investigate any changes in childhood vaccine acceptance with a focus on raising awareness and the implementation of timely and appropriate measures to promptly impede declines in vaccination rates.

## Figures and Tables

**Figure 1 vaccines-10-00814-f001:**
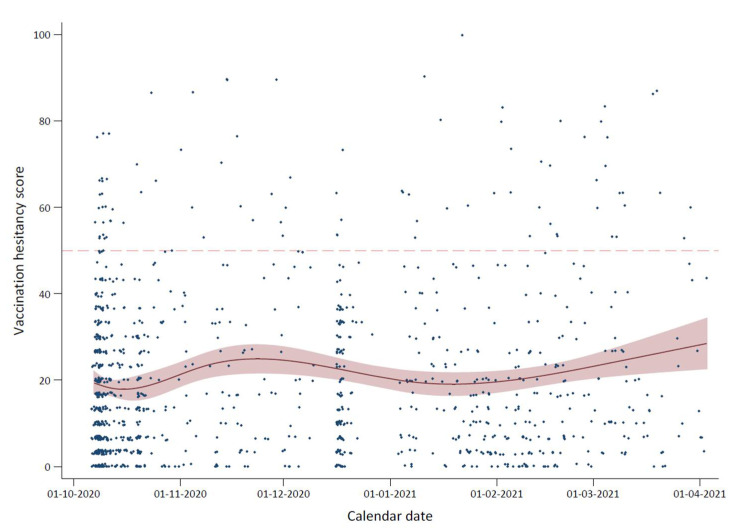
Mean childhood vaccine hesitancy score over time for the 1095 parents who answered the PACV questionnaire. The X-axis represents time from October 2020 to April 2021. The Y-axis represents the continuous vaccine hesitancy score.

**Figure 2 vaccines-10-00814-f002:**
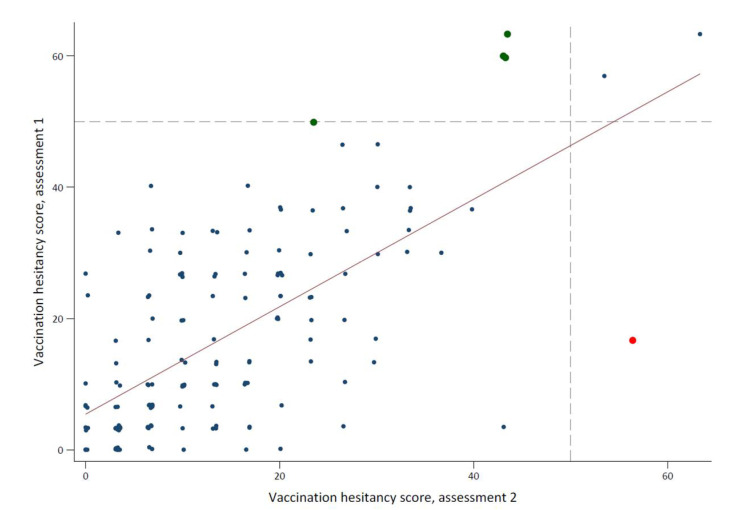
Comparison of childhood vaccine hesitancy score between first assessment and second assessment, for the 156 participants who answered the PACV questionnaire a second time. The Y- and X-axes represent the continuous vaccine hesitancy score from the first and the second assessment, respectively. The horizontal and vertical dashed lines show the cut-off points of vaccine hesitancy.

**Table 1 vaccines-10-00814-t001:** Socio-demographic, lifestyle, and health-related characteristics of participants overall and according to childhood vaccine hesitancy.

	Total N = 1095	Non-Hesitant N = 997 (91.1%)	Hesitant N = 98 (8.9%)	*p*-Value
**Women, N (%)**	715 (65.3)	651 (65.3)	64 (65.3)	>0.99
**Age (years), mean (SD)**	50.25 (9.37)	50.33 (9.32)	49.43 (9.90)	0.364
**BMI (kg/m^2^), mean (SD) ***	26.25 (4.66)	26.26 (4.67)	26.22 (4.55)	0.948
**Total METs [kcal/(kg × h)], mean (SD) ****	21.31 (29.67)	19.97 (26.17)	35.64 (52.56)	5.7 × 10^−6^
**Marital status, N (%)**				
Unmarried/divorced/widowed	155 (14.3)	131 (13.2)	24 (25.5)	
Married/in cohabitation agreement	928 (85.7)	858 (86.8)	70 (74.5)	0.001
**Education, N (%)**				
Up to high school	278 (25.5)	243 (24.4)	35 (36.8)	
University degree	458 (42.0)	409 (41.1)	49 (51.6)	
Master's degree or higher	354 (32.5)	343 (34.5)	11 (11.6)	2.1 × 10^−5^
**Profession, N (%)**				
Executive/scientist/artist/office	769 (76.1)	722 (78.1)	47 (54.7)	
Providing services/seller	99 (9.8)	88 (9.5)	11 (12.8)	
Manual labor	142 (14.1)	114 (12.3)	28 (32.6)	4.1 × 10^−7^
**Income (Euros/month), N (%)**				
≤900	270 (27.3)	226 (25.0)	44 (51.8)	
901–1100	177 (17.9)	167 (18.5)	10 (11.8)	
1101–1400	293 (29.6)	277 (30.6)	16 (18.8)	
>1400	249 (25.2)	234 (25.9)	15 (17.6)	3.5 × 10^−6^
**Smoking status, N (%)**				
Non-smokers	414 (37.9)	384 (38.6)	30 (30.9)	
Former smokers	256 (23.5)	240 (24.1)	16 (16.5)	
Current smokers	421 (38.6)	370 (37.2)	51 (52.6)	0.010
**Alcohol consumption, N (%)**				
Never	172 (15.8)	158 (15.9)	14 (14.3)	
Less than once/month	318 (29.1)	289 (29.1)	29 (29.6)	
1–3 times/month	253 (23.2)	226 (22.7)	27 (27.6)	
1–2 times/week	234 (21.4)	218 (21.9)	16 (16.3)	
3 or more times/week	115 (10.5)	103 (10.4)	12 (12.2)	0.617
**Weight change during last 6 months, N (%)**				
No change	574 (52.8)	518 (52.3)	56 (57.1)	
Lost weight	196 (18.0)	182 (18.4)	14 (14.3)	
Gained weight	318 (29.2)	290 (29.3)	28 (28.6)	0.538
**Self-perceived health status, N (%)**				
Moderate/bad/very bad	152 (13.9)	138 (13.9)	14 (14.4)	
Good	573 (52.5)	536 (53.9)	37 (38.1)	
Very good	366 (33.5)	320 (32.2)	46 (47.4)	0.006
**Restriction of activities during last year due to health issue, N (%)**				
None	622 (57.5)	552 (56.1)	70 (71.4)	
A little/moderately	291 (26.9)	274 (27.8)	17 (17.3)	
Much/very much	169 (15.6)	158 (16.1)	11 (11.2)	0.013
**Stress symptoms during last 2 weeks, N (%)**				
No days	192 (17.5)	173 (17.4)	19 (19.4)	
Some days	607 (55.4)	566 (56.8)	41 (41.8)	
More than half days	144 (13.2)	131 (13.1)	13 (13.3)	
Almost every day	152 (13.9)	127 (12.7)	25 (25.5)	0.003
**Depressive symptoms during last 2 weeks, N (%)**				
No days	374 (34.2)	347 (34.8)	27 (27.6)	
Some days	516 (47.1)	479 (48.0)	37 (37.8)	
More than half days	122 (11.1)	104 (10.4)	18 (18.4)	
Almost every day	83 (7.6)	67 (6.7)	16 (16.3)	2.4 × 10^−4^
**Self-reported presence of chronic disease, N (%)**	392 (36.5)	365 (37.2)	27 (28.7)	0.101
**Physician diagnosis of diabetes, N (%)**	79 (7.2)	75 (7.5)	4 (4.1)	0.208
**Physician diagnosis of high cholesterol, N (%)**	417 (38.3)	398 (40.1)	19 (19.4)	5.6 × 10^−5^
**Physician diagnosis of high blood pressure, N (%)**	237 (21.7)	225 (22.6)	12 (12.4)	0.019
**Area of residence, N (%)**				
Attica	210 (19.3)	186 (18.8)	24 (24.5)	
Peloponnese	48 (4.4)	40 (4.0)	8 (8.2)	
Islands (Aegean and Ionian, Crete)	39 (3.6)	34 (3.4)	5 (5.1)	
Thessaly and Central Greece	48 (4.4)	44 (4.4)	4 (4.1)	
Epirus	612 (56.3)	574 (58.0)	38 (38.8)	
Macedonia and Thrace	131 (12.0)	112 (11.3)	19 (19.4)	0.006

**Abbreviations:** BMI, body mass index; MET, metabolic equivalent of (physical activity) tasks; and SD, standard deviation. * Non-hesitant: N = 967; hesitant: N = 91. ** Non-hesitant: N = 848; hesitant: N = 80.

**Table 2 vaccines-10-00814-t002:** Odds ratios (OR) and 95% confidence intervals (CI) for the association between COVID-19-related variables and childhood vaccine hesitancy, using multivariable logistic regression models.

	Minimally Adjusted Model ^1^		Maximally Adjusted Model ^2^	
	OR (95% CI)	*p*-Value	OR (95% CI)	*p*-Value
**Coronavirus existence**				
Definitely not/probably not/do not know	Ref		Ref	
Probably yes	0.21 (0.09, 0.50)		0.20 (0.05, 0.82)	
Definitely yes	0.05 (0.02, 0.12)	2.1 × 10^−13^	0.05 (0.01, 0.19)	7.8 × 10^−7^
**Knowledge about COVID-19**				
Poor knowledge	Ref		Ref	
Moderate knowledge	0.64 (0.31, 1.30)		0.48 (0.18, 1.31)	
Good knowledge	0.45 (0.27, 0.75)	0.009	0.49 (0.25, 0.96)	0.089 *
**Following COVID-19 measures of personal protection** ^3,4^				
Never/rarely/sometimes	Ref		Ref	
Frequently	0.22 (0.10, 0.46)		0.33 (0.11, 0.97)	
Almost always/always	0.06 (0.03, 0.11)	7.8 × 10^−16^	0.09 (0.03, 0.22)	2.9 × 10^−7^
**Following COVID-19 measures of public protection** ^3,5^				
Never/rarely/sometimes	Ref		Ref	
Frequently	0.22 (0.09, 0.51)		0.23 (0.07, 0.75)	
Almost always/always	0.06 (0.03, 0.12)	2.8×10^−14^	0.06 (0.02, 0.17)	1.56 × 10^−7^
**Trust in health authorities for minimizing the spread of coronavirus** ^3^				
No trust	Ref		Ref	
Little trust	0.29 (0.15, 0.54)		0.26 (0.11, 0.62)	
Some trust	0.14 (0.08, 0.25)		0.12 (0.05, 0.27)	
Absolute trust	0.05 (0.01, 0.16)	6.0×10^−12^	0.04 (0.01, 0.19)	2.3 × 10^−7^
**Trust in official information for the new pandemic** ^3^				
No trust	Ref		Ref	
Little trust	0.31 (0.17, 0.58)		0.48 (0.21, 1.09)	
Some/absolute trust	0.12 (0.06, 0.23)	1.6 × 10^−10^	0.08 (0.03, 0.23)	4.4 × 10^−6^
**Trust in Government for minimizing the spread of coronavirus** ^3^				
No trust	Ref		Ref	
Little trust	0.32 (0.16, 0.64)		0.46 (0.20, 1.08)	
Some trust	0.21 (0.11, 0.41)		0.18 (0.07, 0.47)	
Absolute trust	0.08 (0.02, 0.34)	1.3 × 10^−7^	0.05 (0.01, 0.35)	2.0 × 10^−4^
**Seasonal flu vaccination this year** ^3^				
No	Ref		Ref	
Yes	0.08 (0.04, 0.17)	1.2 × 10^−11^	0.08 (0.03, 0.21)	7.8 × 10^−7^
**Seasonal flu vaccination last year** ^3^				
No	Ref		Ref	
Yes	0.18 (0.09, 0.37)	3.2 × 10^−6^	0.18 (0.07, 0.46)	4.4 × 10^−4^
**Capability to protect against coronavirus** ^3^				
No/little capability	Ref		Ref	
Moderate capability	1.40 (0.41, 4.77)		1.20 (0.31, 4.71)	
Absolute capability	2.86 (0.84, 9.75)	0.010	1.79 (0.44, 7.36)	0.442 *
**COVID-19 symptoms during last months** ^3,6^				
No	Ref		Ref	
Yes	1.85 (0.78, 4.42)	0.166 *	0.83 (0.20, 3.34)	0.787 *
**COVID-19 tested** ^3^				
No	Ref		Ref	
Yes	0.75 (0.44, 1.25)	0.268 *	1.11 (0.57, 2.15)	0.757 *
**Family member with COVID-19 diagnosis** ^3^				
No	Ref		Ref	
Yes	0.69 (0.24, 2.00)	0.493 *	0.76 (0.20, 2.85)	0.685 *
**Willingness to vaccinate against coronavirus** ^3^				
No	Ref			
Yes	0.01 (0.00, 0.04)	2.3 × 10^−7^	^‡^	
**Willingness to vaccinate their children against coronavirus** ^3^				
No	Ref		Ref	
Yes	0.02 (0.01, 0.08)	9.1 × 10^−8^	0.02 (0.00, 0.12)	8.1 × 10^−5^

**Abbreviations**: COVID-19, coronavirus disease 2019; Ref, reference category. ^1^ Model adjusted for age, sex, education, and income. ^2^ Model adjusted for age, sex, education, income, depressive symptoms, physical activity measured using total metabolic equivalents, profession, health status, smoking status, and body mass index. ^3^ The responses “Do not know/Do not answer” were not taken into account. ^4^ COVID-19 measures of personal protection: mask use, frequent hand washing, and keeping distance. ^5^ COVID-19 measures of public protection: mask use, covering the nose and mouth when coughing or sneezing, avoiding large concentrations, and staying home or informing the authorities when not feeling well. ^6^ Fever, persistent cough, breathing difficulty, loss of smell or taste. * *p*-value not statistically significant after FDR correction for multiple comparisons. ^‡^ Model could not run because the variable predicted failures perfectly.

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
