# Peer review of "Parental Hesitancy towards the Established Childhood Vaccination Programmes in the COVID-19 Era: Assessing the Drivers of a Challenging Public Health Concern"

_vaccines, 2022, doi:10.3390/vaccines10050814_

Round 1

Reviewer 1 Report

The manuscript explores hesitance, based on a non-probabilistic sampling of the adult Greek population. The design is mildly appropriate, some elements are odd, and the analysis is somewhat narrower than I expected. Some minor linguistic edits are needed, and some formatting changes are needed.

The main worry is the sample, which is convenient, and therefore not representative at all. It is hard to even explain the sampling, as it is Internet-based, with the partial involvement of what appears to be a cohort study. Such a sample does not represent any fraction of the population, and therefore, it is actually impossible even to estimate the extent of bias. Some of the respondents are obviously more likely to participate in research and are probably better off, so the study design is a major, well, the greatest flaw of the paper – having only the Internet-based sample would have incurred lesser error. Second major problem is the lack of relevant variables, namely the wider extent of the political orientation, general beliefs, or other lifestyle and attitudes, related to vaccination. This domain is only partly involved, and some of the questions, notably that on smoking, do not really belong into this study. This reflects the lifestyle, but again, it is impossible to say how smoking affects the vaccination hesitance, as it should be viewed as the broader concept of the general and health-related lifestyle. Next, the study does not do a good job in linking to the existing body of research related to hesitance, which has largely moved on from the operative variables and is more concerned with wider concepts. I fear that 9% of hesitant people is actually very good, as the figures are much worse in other countries (https://doi.org/10.2147/JMDH.S347669). There are some statistical issues, namely the need to provide the exact three digit P values. It is bad to dichotomize the variable, you should have measured a gradient – it would have been much better if you did not binarize the hesitance. The logistic regression is great, but not ideal in this case. If feasible, do check with linear regression if the predictor levels are retained – no need to show this in the manuscript, but for the sakes of the review, you might try this to see if you are missing any interesting results.

Reviewer 2 Report

The manuscript “Parental hesitancy towards the established childhood vaccination programmes in the COVID-19 era: assessing the drivers of a challenging public health concern.” has been reviewed.

General comments to authors

Sars CoV-2 causing coronavirus disease(COVID-19) is one of the most serious problems that has affected the entire globe, only comparable to the 1918 pandemic flu. This paper presents a colateral damage that the pandemic situation has enhanced, the decrease in routine vaccine uptake   in children for sevral reasons. One underlying reason being vaccine hesitancy which had already been nominated as one the 10 great Public Health challenges. Pandemic mitigation measures have caused lack of accessability to Health centers, lack of face to face attendance and fear of contagion in healthcare centers. 

The work is well written yet there are some issues for improvement and to be corrected.

No sample size calculation

Methods section

No sample size calculation

 Has no info as to some variables such as those in Table 1:

Metabolic equivalent task

Chronic disease, N (%)   Include what ?

Diabetes diagnosis, N (%)  This would also be included as a chronic disease the same as  hypertension

High cholesterol diagnosis, N (%)

High blood pressure diagnosis, N (%)

 These variables should be explained in the methodology

Comments and corrections  are posted on the pdf file

Reviewer 3 Report

Derdemezis et al. presented their excellent study to explore the association between vaccine hesitancy of parents, their characteristics, and trust authorities. The study design and the analysis are well described. The vaccination hesitancy score was not significantly changed during the study period, which suggests the difficulties in solving the vaccine hesitancy. Improvement in trust to health authorities can be a clue. The future study should include the source of information for the solution, which is a limitation of the current study.

Author Response

Derdemezis et al. presented their excellent study to explore the association between vaccine hesitancy of parents, their characteristics, and trust authorities. The study design and the analysis are well described. The vaccination hesitancy score was not significantly changed during the study period, which suggests the difficulties in solving the vaccine hesitancy. Improvement in trust to health authorities can be a clue. The future study should include the source of information for the solution, which is a limitation of the current study.

We thank the reviewer for the positive comments and we agree that perhaps future studies should seek to further explore this complex matter, both in the context of COVID-19 and towards finding the key/s to enhancing trust to science and health authorities.

Round 2

Reviewer 1 Report

The authors have appropriatelly improved the manuscript